# Antimicrobial and Defense Proteins in Chronic Rhinosinusitis with Nasal Polyps

**DOI:** 10.3390/medicina59071259

**Published:** 2023-07-06

**Authors:** Rudolfs Janis Viksne, Gunta Sumeraga, Mara Pilmane

**Affiliations:** 1Department of Otorhinolaryngology, Riga Stradins University, Pilsonu Street 13, LV-1002 Riga, Latvia; gunta.sumeraga@rsu.lv; 2Daugavpils Regional Hospital, Vasarnicu Street 20, LV-5417 Daugavpils, Latvia; 3Pauls Stradins Clinical University Hospital, Pilsonu Street 13, LV-1002 Riga, Latvia; 4Institute of Anatomy and Anthropology, Riga Stradins University, Kronvalda Boulevard 9, LV-1010 Riga, Latvia; mara.pilmane@rsu.lv

**Keywords:** rhinosinusitis, nasal polyps, defensins, cathelicidins

## Abstract

*Background and Objectives*: Chronic rhinosinusitis with nasal polyps (CRSwNP) presently remains a difficult disease to manage. Antimicrobial and defense proteins are important factors that could help characterize the role of microorganisms in CRSwNP pathogenesis, as the concept of microbial dysbiosis in CRS is still being considered. Our aim is to investigate the complex appearance, relative distribution and interlinks of human β defensin 2 (HBD-2), human β defensin 3 (HBD-3), human β defensin 4 (HBD-4), and cathelicidin LL 37 (LL 37) in chronic rhinosinusitis with nasal polyps (CRSwNP)-affected human nasal mucosa. *Materials and Methods*: The study group consisted of 48 samples from patients with CRSwNP. Samples were collected during functional endoscopic sinus surgery. The control group consisted of 17 normal healthy nasal mucosa samples gathered during routine septoplasty. β-defensin-2, β-defensin-3, β-defensin-4 and cathelicidin LL 37 in tissue were detected via immunohistochemical analysis. *Results*: HBD-2, HBD-3 and LL 37 were significantly decreased in epithelial cells in both primary and recurrent nasal polyp samples (*p* < 0.001) in comparison to control samples. HBD-2 was decreased in the subepithelial connective tissue of primary nasal polyp samples when compared to both recurrent polyp (*p* = 0.050) and control (*p* = 0.033) samples. In subepithelial connective tissue, significantly more HBD-3-positive structures were observed in primary nasal polyp samples (*p* = 0.049) than in control samples. In primary polyp samples, moderate correlations between connective tissue HBD-3 and connective (R = 0.584, *p* = 0.001) and epithelial tissue LL 37 (R = 0.556, *p* = 0.002) were observed. *Conclusions*: Decreased HBD-2, HBD-3 and LL 37 concentrations in the epithelium suggest a dysfunction of the epithelial barrier in patients with nasal polyps. Decreased subepithelial connective tissue HBD-2 suggests different responses to nasal microbiota in patients with primary nasal polyps compared to recurrent nasal polyps. Increased HBD-3 in subepithelial connective tissue suggests a possible role of this antimicrobial peptide in the pathogenesis of primary nasal polyps.

## 1. Introduction

Chronic rhinosinusitis (CRS) is described as an inflammation of the nasal cavity and paranasal sinuses, with symptoms such as nasal congestion, nasal discharge, and hyposmia/anosmia as well as facial pressure/pain. In the case of CRS, symptoms persist for longer than 12 weeks and cause a significant decrease in quality of life [1,2]. The overall prevalence of CRS, based on diagnostic criteria, is estimated to be around 5%, a third of whom also have nasal polyps present [3]. The presence of nasal polyps is the basis for the division of CRS into two categories: chronic rhinosinusitis with (CRSwNP) and without (CRSsNP) nasal polyps [2]. The presence of nasal polyps indicates more significant morbidity and decreased quality of life [1].

The involvement of bacterial pathogens in the pathogenesis of CRS has been considered. Alteration in the local composition of microorganisms in a sinonasal setting accompanied by disturbance of mucosa is behind the so-called dysbiosis theory [4]. Although no consistent patterns that would implicate a specific microorganism as a cause for CRS have been found, the concepts of local inflammation caused by microbiome imbalance and immunomodulatory properties achieved by the correct balance of local microbes are still being considered [5]. Nasal secretions contain bacteria even without the presence of acute or chronic infections. Specific and nonspecific mechanisms act to protect nasal tissue from infection, such as macrophages, neutrophils, mucociliary clearance and antimicrobial proteins secreted by nasal tissue [6]. Antimicrobial peptides are described as the new antibiotics. They possess antimicrobial as well as immunoregulatory, anti-biofilm and other activities [7]. They are found in almost all life forms. These molecules, also known as host defense peptides, have an important role in innate immune reactions with the fundamental purpose of protecting cells from bacterial and fungal infections. In vertebrate organisms they can be found in white blood cells and epithelial surface cells in such places as the oropharyngeal region, lungs, and nasal cavity. Defensins and cathelicidins are the usual groups of antimicrobial peptides found in vertebrates [8].

Defensins are cationic peptides containing 4–5 disulphide bridges with a size of 45 to 54 amino acids [9]. The defensins found in the human body are human beta defensins 1–4 (HBD 1–4). HBD-1 is expressed constantly with the possibility of being upregulated, while HBD 2–4 expression is induced as a response to pro-inflammatory stimuli [10].

Human beta defensin 2 (HBD-2) is secreted by epithelial cells when encountering microorganisms or by activation of pro-inflammatory cytokines. Pro-inflammatory cytokines or pathogen-associated molecular patterns can activate various signaling pathways via toll-like receptors 2 and 4. Along with antimicrobial activity, HBD-2 can also stimulate toll-like receptor expression on the surface of immune cells as well as their chemotaxis. HBD-2 has an important defensive role in the oral cavity. Gingival tissue has demonstrated continuous secretion of HBD-2 in the absence of inflammation, but oral keratinocytes produce it only when stimulated by proinflammatory cytokines in response to bacterial exotoxins [11]. HBD-2 has not been thoroughly researched in nasal mucosa. One study observed HBD-1 and HBD-2 presences in the nasal epithelium in both normal and inflamed sinuses using immunostaining, with increased secretion concentrations during inflammation. Deficiency in these peptides as a possible cause for chronic rhinosinusitis was proposed [6]. Another study found increased levels of HBD-2 in samples from nasal polyps when compared to control samples from normal nasal mucosa. The PCR method was used to estimate expression of the HBD-2 gene. The biotin–streptavidin immunohistochemical method was used to detect only HBD-2 and its localization with no additional quantitative measurements [12]. Increased HBD-2 levels after endoscopic sinus surgery have been shown to predict better response after surgical treatment [13]. Increased HBD-2 levels have been observed in the respiratory tract tissue of smokers, suggesting a role of this protein in protecting the respiratory tract against oxidative stress and related infections [11].

Human beta defensin 3 (HBD-3) also exhibits antimicrobial activity as well as adaptive immune reactions such as chemotaxis. HBD-3 has shown broad spectrum activity against several drug-resistant bacteria [10]. This peptide is found in the epithelial cells of the respiratory, urogenital, and gastrointestinal tracts as well as in non-epithelial tissues such as tonsils and muscular tissue. Similarly, as in the case of HBD-2, HBD-3 expression is also induced by bacteria, TNF-α and IL-1β [14]. HBD-3 has demonstrated effectiveness in elimination of biofilms of *Staphylococcus* antibiotic-resistant strains [15]. HBD-3-class peptides are considered to be a possible novel antimicrobial therapeutic option to limit modern antibiotic overuse [16]. However, *Staphylococcus aureus* has been shown to be capable of adapting its phenotype to become resistant to both HBD-2 and HBD-3 in cystic fibrosis patients [17]. HBD-3 has shown strong activity against Gram-negative bacteria such as *Pseudomonas aeruginosa*, *Acinetobacter baumannii* and *Stenotrophomonas maltophilia* with bactericidal action within 1–5 min. HBD-3 shows slower activity against Gram-positive species, from 10 to 20 min. Salt resistance has been observed in HBD-3, giving it strong antibacterial activity and the ability to retain physiological salt concentrations [18].

Human beta defensin 4 (HBD-4) has been observed in the testis, uterus, kidneys, and gastric mucosa. HBD-4 expression is also seen in human respiratory mucosa in response to *P. aeruginosa* and *S. pneumoniae*. HBD-3 and HBD-4 have both demonstrated the ability to activate mast cells and increase vascular permeability [14]. HBD-4 possesses the strongest activity against *P. aeruginosa* out of all human beta defensins [19]. HBD-3 and HBD-4 are characterized by synergistic behavior in healthy human lung tissue [20].

Cathelicidin LL 37 (LL 37) is a mature form of human cathelicidin. It is named after its possibility to inhibit the protease cathepsin-L. LL 37 serves as a primary defense against bacteria and other pathogens during inflammation. Cathelicidins can destroy biofilms, viruses, parasites, and fungi and modulate as well as stimulate cells of the innate and adaptive immune systems. LL 37 is expressed in various tissues such as the intestinal epithelium, respiratory system, and genitals. Cathelicidins can also play a role in angiogenesis, wound healing and apoptosis regulation [21]. LL 37 has been more thoroughly described in association with chronic rhinosinusitis than defensins. Its expression is observed in normal human nasal mucosa and becomes significantly upregulated during inflammation [22]. LL 37 has been shown to promote the formation of neutrophil extracellular traps in CRSwNP [23]. LL 37 also seems to be upregulated in CRS patients in response to fungal pathogens [24].

The previously described antimicrobial proteins possess activity against microorganisms found in the nasal cavity and paranasal sinuses; therefore, our aim is to investigate the complex appearance, relative distribution and interlinks of human β defensin 2 (HBD-2), human β defensin 3 (HBD-3), human β defensin 4 (HBD-4), and cathelicidin LL 37 (LL 37) in chronic rhinosinusitis with nasal polyps (CRSwNP)-affected human nasal mucosa.

## 2. Materials and Methods

A total of 48 patients with previous diagnosis of CRSwNP and no known immunodeficiencies or coagulopathies were included in the study group. Patients with CRS exacerbation less than two weeks prior to surgery were excluded. Nasal polyp samples were taken during functional endoscopic sinus surgery (FESS). The study group was further divided into 29 patients undergoing their first surgery (patients with primary nasal polyps), 29 to 78 years old, with an average age of 48 years and 19 patients experiencing a recurrence of nasal polyps after previous surgeries (patients with recurrent nasal polyps), 31 to 88 years old, with an average age of 56 years. The control group consisted of 17 otherwise healthy individuals, with an age range from 29 to 74 years and an average of 39 years, with isolated nasal septum deviation and no known diagnosis of CRS or any other sinonasal pathologies. Control group patients had no history of nasal surgery. Patients with immunodeficiencies and coagulopathies were excluded from the study group. Mucosa samples were taken from mucosa of inferior nasal turbinates. The nature of the study was explained, and written consent was obtained from each participant. The study was carried out with the approval of the local ethics committee of Riga Stradins University (6-1/10/59. 26 October 2020).

Samples were collected and placed in a mixture of 2% formaldehyde and 0.2% picric acid in 0.1 M phosphate buffer, with a pH of 7.2, for up to 72 h; afterwards, they were rinsed in Tyrode’s buffer (NaCl, KCl, CaCl_2_·2H_2_O, MgCl_2_·6H_2_O, NaHCO_3_, NaH_2_PO_4_·H_2_O, and glucose) containing 10% saccharose for 12 h and then embedded into paraffin. Three-micrometre-thin sections were cut and biotin–streptavidin immunohistochemical staining was used to detect HBD-2 (sc-20798, working dilution 1:100, Santa Cruz Biotechnology, Inc., Dallas, TX, USA); HBD-4 (ab14419, working dilution 1:200, Abcam, San Francisco, CA, USA); HBD-3 (rb183268, working dilution 1:100, Biorbyt Limited, Cambridge, UK); LL 37 (orb88370, working dilution 1:100, Biorbyt Limited, Cambridge, UK).

Light microscopy at ×200 magnification was used to analyze the number of positive structures in stained samples. When evaluating immunohistochemical results, researchers commonly assess the relative percentage of positively stained cells in relation to the total number of target cells. For example, each value could be recorded as a number score representing every 10% (0, 0–9%; 1, 10–19%; 2, 20–29%; 3, 30–39%; 4, 40–49%; 5, 50–59%; 6, 60–69%; 7, 70–79%; 8, 80–89%; 9, 90–100%) [25]. It is accepted to transform percentages into “+” values and numbers for statistical purposes. The semi-quantitative method and scoring system levels used in this research are well established and were widely used in previous morphological studies for several decades [26,27], and are still used and accepted in recent publications [28,29]. The appearance of positive structures was recorded using a semi-quantitative method where: 0—no positive structures in the visual field; 0/+—occasional positive structures in the visual field; +—few positive structures; ++—a moderate number of positive structures; +++—numerous positive structures; ++++—abundant positive structures in the visual field [20]. Results were transformed into numerical values—0/+ to 0.5, + to 1, ++ to 2, +++ to 3 and ++++ to 4—for statistical analysis. If expressed in percentages, our values correspond in the following manner: + or 1, 0–24%; ++ or 2, 25–49%; +++ or 3, 50–74% and ++++ or 4, 75–100%. Statistical methods such as Spearman’s rank correlation and the Mann–Whitney U test were used for analysis.

## 3. Results

HBD-2 was significantly decreased in epithelial cells in both primary (*p* < 0.001) and recurrent nasal polyp samples (*p* < 0.001) when compared to controls. HBD-2 was decreased in the subepithelial connective tissue of primary nasal polyp samples when compared to both recurrent polyp (*p* = 0.050) and control (*p* = 0.026) samples. In addition, HBD-3 was observed to be significantly decreased in the epithelia of both primary (*p* < 0.001) and recurrent (*p* < 0.001) nasal polyp samples in comparison to the epithelial cells of control samples. In subepithelial connective tissue, significantly more HBD-3-positive structures were observed in primary nasal polyp samples (*p* = 0.049) than in control samples. Significantly decreased LL 37 was seen in both primary (*p* < 0.001) and recurrent nasal polyp samples (*p* < 0.001) after comparison with the control group (Table 1).

An occasional presence of HBD-2-positive structures (0/+) was observed in the epithelia of both primary and recurrent nasal polyp samples, but in subepithelial connective tissue, primary polyp samples mainly showed few positive structures (+) while recurrent polyp samples had a few to moderate (+/++) number of positive structures (Figure 1a). Control samples, on the other hand, had numerous HBD-2-positive structures in epithelial cells (+++) and few to moderate (+/++) positive structures in subepithelial connective tissue (Figure 1b).

HBD-3 showed occasional positive structures (0/+) in both primary and recurrent nasal polyp epithelia, and both showed few to moderate positive structures (+/++) in subepithelial connective tissue (Figure 1c). Control samples had a moderate number of HBD-3-positive structures in the epithelium (++) and few (+) in subepithelial connective tissue (Figure 1d).

HBD-4-positive structures were not detected in nasal polyp tissue (Figure 2a) nor in control sample tissue (Figure 2b). In primary nasal polyp samples, occasional LL 37-positive structures (0/+) were seen in the epithelium and a few to moderate number (+/++) in subepithelial connective tissue (Figure 2c). In control samples, moderate (++) numbers of LL 37-positive structures were seen in epithelial cells and few to moderate (+/++) were observed in subepithelial connective tissue (Figure 2d).

Correlations between factors in primary and recurrent nasal polyp samples were evaluated using Spearman’s rank correlation. Values of coefficients: ≥0.81—a very strong correlation, 0.61–0.80—a strong correlation, 0.41–0.60—moderate correlation, 0.21–0.40—weak correlation, 0.01–0.20—no correlation between two factors.

When analyzing primary nasal polyp samples, a strong correlation was observed between connective tissue LL 37 and epithelial LL 37 (R = 0.758, *p* < 0.001). Moderate correlations were seen between connective tissue LL 37 and connective tissue HBD-3 (R = 0.584, *p* = 0.001), connective tissue HBD-2 and epithelial HBD-2 (R = 0.561, *p* = 0.002) and epithelial LL 37 and connective tissue HBD-3 (R = 0.556, *p* = 0.002) (Table 2).

When recurrent polyp samples were evaluated, a strong correlation between epithelial HBD-2 and connective tissue HBD-2 (R = 0.635, *p* = 0.004) and a moderate correlation between epithelial HBD-2 and epithelial HBD-3 (R = 0.505, *p* = 0.027) were found (Table 3).

## 4. Discussion

Our results demonstrate significantly decreased levels of HBD-2, HBD-3 and LL 37 in the epithelial cells of nasal polyps in contrast to normal nasal mucosa. This is somewhat similar to our previous research on cytokines in CRSwNP; cytokines were also shown to be decreased in the epithelium of nasal polyps in contrast to normal nasal mucosa [30]. This further leads us to consider a dysfunctional epithelial barrier and epithelial cell hypofunction as important factors in the genesis of CRSwNP. It has been previously described that defects in the function of the epithelial barrier and loss of its integrity, combined with colonization by microorganisms and lack of expression of antimicrobial products, play an important role in the pathogenesis of nasal polyps [31]. It has been observed that *Staphylococcus aureus* could affect the function of the epithelial barrier in nasal mucosa; in fact, enterotoxin B could be a driving factor in CRS exacerbation [32]. Furthermore, restoration of epithelial barrier function is considered to be a novel approach in treatment of CRS [33].

Our findings suggest an increased role of HBD-3 in patients with primary nasal polyps due to the elevated HBD-3 concentration in subepithelial connective tissue in comparison to control samples and recurrent nasal polyp samples. As stated before, HBD-3 has shown effectiveness in eliminating biofilms of *Staphylococcus* antibiotic-resistant strains [15]. Research has suggested that in patients with recurring nasal polyps and repeated nasal surgeries, biofilms are more likely to be present, while with patients naive to surgical interventions, biofilm-producing bacteria are found in about 50% fewer cases [34]. Therefore, the absence of an increased HBD-3 concentration might be one of the facilitating factors in biofilm formation in patients with recurring nasal polyps. The significance of HBD-3 has not been extensively studied in patients with chronic rhinosinusitis. One study attempted to draw conclusions about possible HBD-3 level changes in the nasal secretions of patients with nasal polyps with or without *S. aureus* colonization, but no differences were found [35]. Another study suggests that HBD-3 secretion is greatly reduced in samples of nasal polyps and positive *S. aureus* carriage [36]. *S. aureus* has been detected in CRSwNP tissue without a corresponding increase in surrounding eosinophils, lymphocytes, or neutrophils, despite CRS treatment [37]. In another study, nasal polyp samples showed decreased epithelial barrier expression at tight junctions facilitated by *Staphylococcus aureus*. Interestingly, in healthy nasal mucosa, *Staphylococcus aureus* seems to have an opposite effect [38]. A study of endotypes of CRS found one cluster of patients, with various cytokines, inflammation markers and clinically severe symptoms such as asthma and nasal polyposis, to have increased *Staphylococcus aureus* enterotoxin expression [39]. Initial increases in HBD-3 in primary nasal polyps could characterize an active fight against *S. aureus* subepithelial tissue invasion, and the absence of this HBD-3 increase in recurrent polyp samples could show a failure to stop this process.

The increased formation of biofilms might be associated with dysfunctional mucociliary clearance, as it is observed that mucociliary clearance indeed decreases in patients with nasal polyps [40]. Repeated surgeries might facilitate the formation of biofilms even further due to changes in anatomy or scarring after surgery. This also raises a question—is biofilm formation associated with bacterial factors and decreased antimicrobial protein (especially HBD-3) activity, or are the bacteria more easily colonizing dysfunctional, scarred tissue that formed after numerous surgeries? Biofilm formation and nasal discharge stasis have been addressed as important factors in patient postoperative care [41].

We found decreased HBD-2 concentrations in subepithelial tissue in patients with primary nasal polyps when compared to both control group patients and recurrent nasal polyp patients. However, in the recurrent nasal polyp group, there was no significant difference compared to the control group. When we compare our findings to previous studies, we can see controversial findings. In one study HBD-2 was expressed in patients with primary and recurrent nasal polyps, while expression was rarely observed in patients with objective recovery and control subjects; this analysis was performed using RT-PCR that detected mRNA [13]. Another study suggests that HBD-2 mediates the early-stage antimicrobial defense in nasal mucosa and that diminished HBD-2 gene expression could explain persistent *Pseudomona aeruginosa* infections [42].

HBD-4-positive structures were not observed in our tissue samples of nasal polyps and normal nasal mucosa. Therefore, HBD-4 does not seem to play an important role in the pathogenesis or sustainment of CRSwNP.

There are not many available studies regarding LL 37′s significance within CRSwNP. It is suggested that LL 37 expression is increased in nasal polyps and therefore a contributing factor to their pathogenesis [43]. Controversially, in a different study, disturbed LL 37 was shown to be a possible cause for dysregulation of the nasal barrier in nasal polyps [35]. Our results show moderate correlations between connective tissue HBD-3 and connective and epithelial tissue LL 37. However, LL 37 had no notable correlations with other factors in samples from recurrent nasal polyps, which also leads us to believe that the antimicrobial peptide response in primary nasal polyps might be more pronounced and coordinated than in recurring polyps.

Discrepancies between our results and those of previous studies could be accounted for by different methods of analyzing tissue samples. For example, an enzyme-linked immunosorbent assay (ELISA) is a protein-based detection method whereas a real-time polymerase chain reaction (PCR) is a deoxyribonucleic acid (DNA)-based detection method [44]. Therefore, different methods, including immunohistochemical staining, could fundamentally yield different interpretations of the underlying mechanisms of CRS. When comparing control samples to nasal polyps, the anatomical sampling site is a possible cause for discussion. As stated before, our control samples were taken from the inferior turbinates of the nasal cavity. Since the respiratory parts of the nasal mucosa are lined with the same type of epithelium and the subepithelial tissue is structurally the same as in the ethmoidal region, mucosa of the inferior nasal turbinates can serve as a reference for normal nasal mucosa [45].

This study possesses various limitations. When describing HBD-2 and HBD-3 in the context of Gram-positive and Gram-negative bacteria, bacterial cultures and analysis of the microbiome and microbiota should be performed. Analysis with ELISA would be beneficial in evaluating antimicrobial peptide concentration. Additional data, such as use of medications and known comorbidities, would be beneficial in future studies.

## 5. Conclusions

Decreased HBD-2, HBD-3 and LL 37 concentrations in the epithelium suggest a dysfunction of the epithelial barrier in patients with nasal polyps. Decreased subepithelial connective tissue HBD-2 suggests different responses to nasal microbiota in patients with primary nasal polyps compared to recurrent nasal polyps. Increased HBD-3 in subepithelial connective tissue suggests a possible role of this antimicrobial peptide in the pathogenesis of primary nasal polyps.

## Figures and Tables

**Figure 1 medicina-59-01259-f001:**
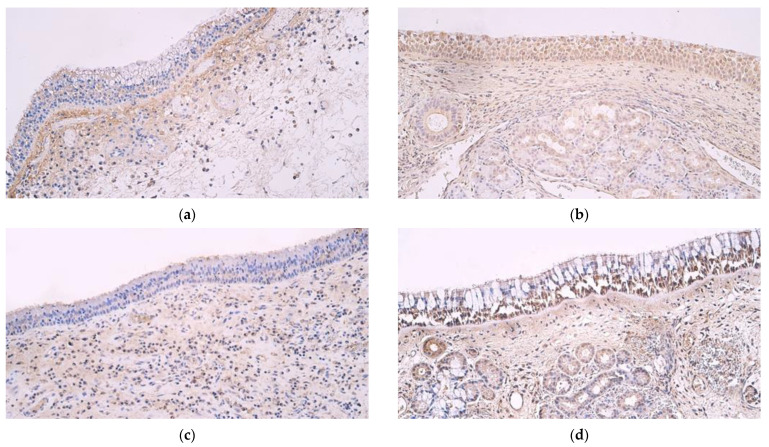
(**a**–**d**) Immunohistochemical micrographs from patients with nasal polyps and control subjects. (**a**) Polyp sample with occasional to few β defensin 2-positive structures in epithelium and a moderate number of positive structures in connective tissue. Β defensin 2 IMH, ×200. (**b**) Note numerous β defensin 2-positive structures in the epithelium as well as few to moderate positive structures in connective tissue in a control sample of normal nasal mucosa. β defensin 2 IMH, ×200. (**c**) Nasal polyp sample with few β defensin 3-positive structures in epithelium but moderate to numerous positive structures in connective tissue. β defensin 3 IMH, ×200. (**d**) Note abundant β defensin 3-positive structures in the epithelium of a control sample and few to moderate positive structures in the connective tissue. β defensin 3 IMH, ×200.

**Figure 2 medicina-59-01259-f002:**
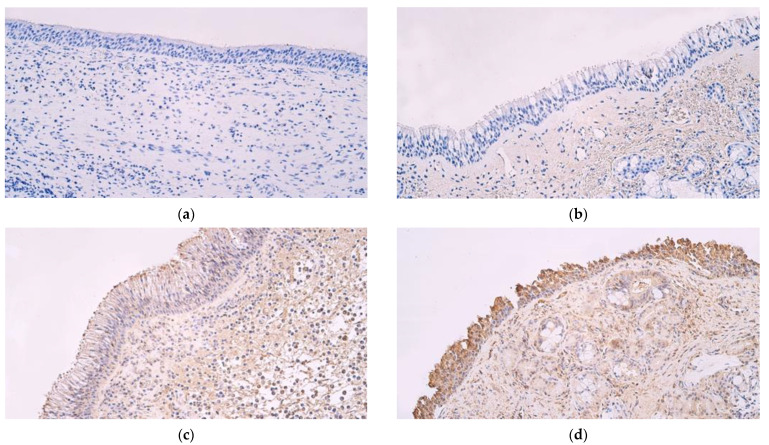
(**a**–**d**) Immunohistochemical micrographs from patients with nasal polyps and control subjects. (**a**) No β defensin 4-positive structures in epithelium or connective tissue of a nasal polyp. β defensin 4 IMH, ×200. (**b**) No β defensin 4-positive structures in a control sample tissue. β defensin 4 IMH, ×200. (**c**) A sample of a nasal polyp showing few to moderate Cathelicidin LL 37-positive structures in the epithelium and numerous positive structures in connective tissue. Cathelicidin LL 37 IMH, ×200. (**d**) Abundant Cathelicidin LL 37-positive structures in the epithelium and few to moderate positive structures in subepithelial connective tissue of a control sample. Cathelicidin LL 37 IMH, ×200.

**Table 1 medicina-59-01259-t001:** Mean number of human β defensin 2-, human β defensin 3-, human β defensin 4- and cathelicidin LL 37-positive structures in samples from nasal polyps and significant differences with the control group.

	Epithelial β Defensin 2	Connective Tissue β Defensin 2	Epithelial β Defensin 3	Connective Tissue β Defensin 3	Epithelial β Defensin 4	Connective Tissue β Defensin 4	Epithelial Cathelicidin LL 37	Connective Tissue Cathelicidin LL 37
Primary polyps	0.328 (SD ± 0.603)	1.052 (SD ± 0.939)	0.603 (SD ± 0.588)	1.552 (SD ± 1.080)	0	0	0.534 (SD ± 0.597)	1.534 (SD ± 1.008)
Recurrent polyps	0.526 (SD ± 0.716)	1.632 (SD ± 0.970)	0.342 (SD ± 0.473)	1.263 (SD ± 1.147)	0	0	0.816 (SD ± 0.691)	1.632 (SD ± 0.831)
Control group	3.059 (SD ± 0.659)	1.647 (SD ± 0.386)	1.853 (SD ± 0.843)	0.912 (SD ± 0.618)	0	0	2.176 (SD ± 0.809)	1.294 (SD ± 0.811)
Primary vs. recurrent group	*p* = 0.283	*p* = 0.050	*p* = 0.096	*p* = 0.387	*p* > 0.999	*p* > 0.999	*p* = 0.145	*p* = 0.568
Primary vs. control group	*p* < 0.001	*p* = 0.026	*p* < 0.001	*p* = 0.049	*p* > 0.999	*p* > 0.999	*p* < 0.001	*p* = 0.610
Recurrent vs. control group	*p* < 0.001	*p* = 0.950	*p* < 0.001	*p* = 0.531	*p* > 0.999	*p* > 0.999	*p* < 0.001	*p* = 0.315

Abbreviations: SD—standard deviation, *p*—*p* value.

**Table 2 medicina-59-01259-t002:** Correlations among factors in primary nasal polyp samples.

Factor 1	Factor 2	R	*p* Value
Connective tissue Cathelicidin LL 37	Epithelial Cathelicidin LL 37	0.758 **	<0.001
Connective tissue Cathelicidin LL 37	Connective tissue β defensin 3	0.584 **	0.001
Connective tissue β defensin 2	Epithelial β defensin 2	0.561 **	0.002
Epithelial Cathelicidin LL 37	Connective tissue β defensin 3	0.556 **	0.002

Abbreviations: R—correlation coefficient; **—correlation is significant at the 0.01 level.

**Table 3 medicina-59-01259-t003:** Correlations among factors in recurrent nasal polyp samples.

Factor 1	Factor 2	R	*p* Value
Epithelial β defensin 2	Connective tissue β defensin 2	0.635 **	0.004
Epithelial β defensin 2	Epithelial β defensin 3	0.505 *	0.027

Abbreviations: R—correlation coefficient; **—correlation is significant at the 0.01 level; *—correlation is significant at the 0.05 level.

## Data Availability

The data presented in this study are available upon request from the corresponding author.

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
