# Peer review of "Antimicrobial and Defense Proteins in Chronic Rhinosinusitis with Nasal Polyps"

_medicina, 2023, doi:10.3390/medicina59071259_

Round 1

Reviewer 1 Report

Summary

In this manuscript, the authors examined the expression of antimicrobial peptides, including HBD-2, HBD-3, HBD-4, and LL-37 in sinonasal tissue of control subjects and patients with CRSwNP by performing immunohistochemistry analyses. They found that the expression of HBD-2, HBD-3 and LL-37 was significantly decreased in epithelial cells in both primary and recurrent nasal polyp samples compared to control samples. In addition, they observed a decreased HBD-2 expression combined with an increased HBD-3 expression in sub-epithelial connective tissue in patients with CRSwNP compared to control subjects. Collectively, the authors suggested different tissue responses to nasal microbiota in patients with primary nasal polyps compared to recurrent nasal polyps.

General comments

In general, the study is descriptive and ambiguous descriptions on the quantification method weaken the rigor of this study. In addition, poor presentation and description of the data lower the priority and novelty of the manuscript. Furthermore, the conclusions are incorrect and overstatement.

Specific points

1.       My major concern is the quantification method of staining results. What were the criteria for differentiating between “numerous”, “moderate”, “few”, and “occasional”? Was it subjective? Who did quantify? Please provide the more detailed information on criteria for the quantification and methods (e.g. the number or percentage of positive structures; the number of slides reviewed; magnification, etc.).

2.       Table 1: The authors described table 1 as the mean number of structures in samples. However, the data presented in table 1 do not seem the average number. The authors described in the Methods section that results were transferred to numerical values: 0/+ to 0.5; + to 1; ++ to 2; +++ 154 to 3 and ++++ to 4 for statistical analysis. In addition to p-value, please describe the scores (mean and standard deviation) of each patient group regarding each parameter.

3.       Please provide the number of samples in “primary” and “recurrent” nasal polyps. This is the basic requirement for scientific writing in biomedical research field.

4.       The conclusion “Decrease of HBD-2 combined with an increase of HBD-3 in subepithelial connective tissue suggests different tissue responses to nasal microbiota in patients with primary nasal polyps compared to recurrent nasal polyps” seems incorrect. There was no significant difference in HBD-3 expression between primary nasal polyps and recurrent nasal polyps (Table 1).

5.       The reason for discrepancy in the results compared to previous studies should be discussed.

6.       How about the study participants’ history of antibiotics or steroid use?

7.       Please describe the anatomical site (inferior turbinate, ethmoid sinus mucosa, etc.) of tissues obtained in the control group.

8.       It is well-known that inflammatory microenvironment and pathophysiology of CRS are highly heterogeneous according to the inflammatory endotypes. How about the distribution of CRS endotype in the study subjects?

9.       Please provide the clinical and laboratory characteristics of the study subjects, such as asthma comorbidity, blood eosinophil count/percentage, serum IgE, and allergic status.

10.    Terminology should be consistent within the manuscript (e.g. LL-37, LL37, LL 37).

Author Response

Reviewer 1

Thank you for your valuable and constructive criticism. We addressed Your comments and suggestions point by point. Changes in the body of the manuscript are marked in green.

  1. My major concern is the quantification method of staining results. What were the criteria for differentiating between “numerous”, “moderate”, “few”, and “occasional”? Was it subjective? Who did quantify? Please provide the more detailed information on criteria for the quantification and methods (e.g. the number or percentage of positive structures; the number of slides reviewed; magnification, etc.).
  • A more detailed explanation of the methods was added to the manuscript:
    Light microscopy with x200 magnification was used to analyze the number of positive structures in stained samples. In evaluating immunohistochemical results, researchers commonly assess the relative percentage of positively stained cells in relation to the total number of target cells. For example, each value could be recorded as a number score in every 10% (0, 0%–9%; 1, 10%–19%; 2, 20%–29%; 3, 30%–39%; 4, 40%–49%; 5, 50%–59%; 6, 60%–69%; 7, 70%–79%; 8, 80%–89%; and 9, 90%–100%) [25]. It is accepted to transfer percentages to ‘’+’’ values and numbers for statistical purposes. Semi-quantitative method and scoring system levels used in this research are well established and widely used in previous morphological studies for several decades [26], [27] and are still used and accepted in recent publications [28], [29]. The appearance of positive structures was recorded by using semi-quantitative method where: 0 - no positive structures in the visual field; 0/+ - occasional positive structures in the visual field; + - few positive structures; ++ - moderate number of positive structures; +++ - numerous positive structures; ++++ - abundant positive structures in the visual field [20]. Results were transferred to numerical values: 0/+ to 0.5; + to 1; ++ to 2; +++ to 3 and ++++ to 4 for statistical analysis. If expressed in percentage, our values would correspond in the following manner: + or 1, 0%-24%; ++ or 2, 25%-49%; +++ or 3, 50%-74% and ++++ or 4, 75%-100%. Statistical methods such as Spearman’s rank correlation and The Mann–Whitney U test were used.

  1. Table 1: The authors described table 1 as the mean number of structures in samples. However, the data presented in table 1 do not seem the average number. The authors described in the Methods section that results were transferred to numerical values: 0/+ to 0.5; + to 1; ++ to 2; +++ 154 to 3 and ++++ to 4 for statistical analysis. In addition to p-value, please describe the scores (mean and standard deviation) of each patient group regarding each parameter.
  • As per Your recommendation the table 1 has been changed to depict mean values and standard deviations of numerical values. In addition, p values also have been provided.
  1. Please provide the number of samples in “primary” and “recurrent” nasal polyps. This is the basic requirement for scientific writing in biomedical research field.
  • Thank you, indeed the number of samples was missing. We have corrected this error and added additional information about patients:
    The study group was further divided into 29 patients undergoing their first surgery (patients with primary nasal polyps), 29 to 78 years old with average age of 48 years and 19 patients experiencing a recurrence of nasal polyps after previous surgeries (patients with recurrent nasal polyps), 31 to 88 years old with average age of 56 years.
  1. The conclusion “Decrease of HBD-2 combined with an increase of HBD-3 in subepithelial connective tissue suggests different tissue responses to nasal microbiota in patients with primary nasal polyps compared to recurrent nasal polyps” seems incorrect. There was no significant difference in HBD-3 expression between primary nasal polyps and recurrent nasal polyps (Table 1).
  • Indeed, there was no significant difference between primary and recurrent nasal polyps in HBD-3 subepithelial connective tissue expression. Yet there was a significant difference between expression of HBD-3 in subepithelial connective tissue when comparing primary nasal polyp samples and the control group. The conclusion was driven by the latter.
  1. The reason for discrepancy in the results compared to previous studies should be discussed.
  • We have added the mentioned pert to the discussion:
    Discrepancy in our results when compared to previous studies could be accounted for by different methods of analyzing tissue samples. For example, enzyme-linked immunosorbent assay (ELISA) is a protein-based detection method whereas real-time polymerase chain reaction (PCR) is a deoxyribonucleic acid (DNA) based detection method [44]. Therefore, different methods, including immunohistochemical staining, fundamentally could yield a different interpretation of the underlying mechanisms of CRS.
  1. How about the study participants’ history of antibiotics or steroid use?
  • This was not the aim of our study, as for now we only seek to characterize morphological changes. The reaction of tissue to irritation is not affected by medication. We have added it as a limitation of the study:
    This study possesses various limitations. When describing HBD-2 and HBD-3 in the context of Gram-positive and Gram-negative bacteria, bacterial cultures, analysis of microbiome and microbiota should be performed. Analysis with ELISA method would be beneficial in evaluating antimicrobial peptide concentration. Additional data such as use of medication and known comorbidities would be beneficial in future studies.
  1. Please describe the anatomical site (inferior turbinate, ethmoid sinus mucosa, etc.) of tissues obtained in the control group.
  • The anatomical sites are described in ‘’materials and methods’’ section:
    Nasal polyp samples were taken during functional endoscopic sinus surgery (FESS).
    and
    Mucosa samples were taken from mucosa of inferior nasal turbinates.
  1. It is well-known that inflammatory microenvironment and pathophysiology of CRS are highly heterogeneous according to the inflammatory endotypes. How about the distribution of CRS endotype in the study subjects?
  • Characterization of possible different CRS endotypes is the future goal of our research in general. In this specific study however, we have no additional data about conventional CRS endotype distribution.
  1. Please provide the clinical and laboratory characteristics of the study subjects, such as asthma comorbidity, blood eosinophil count/percentage, serum IgE, and allergic status.
  • As mentioned before, this was not the aim of our study, but we addressed it in the limitations of the study and hope to include these parameters in future research.
  1. Terminology should be consistent within the manuscript (e.g. LL-37, LL37, LL 37).
  • Indeed, this is corrected throughout the article, thank you.

Reviewer 2 Report

In their research article "Antimicrobial and defense proteins in chronic rhinosinusitis with nasal polyps" the authors analyze samples from 48 CRS patients that underwent surgery for nasal polyps, either primary or recurring. The control group was 17 patients that had routine septoplasty done. The authors then performed immunohistochemical staining to determine the expression of  HBD-2, -3 -4 and LL-37 in the mucosa. They found out that HBD-2, HBD-3 and LL-37 expression was decreased in the epithelium of both patient groups (primary and recurrent surgery), while HBD-2 expression in the connective tissue was comparable to the control group in recurrent polyps. HBD-3 was increased in the connective tissue of primary polyp group. The conclusions drawn from the results are suitable: they are speculative and reflect the fact that the only immunohistochemical staining of the abovementioned proteins was performed and then different patient groups compared, which does not prove any causality or mechanism as such. It is reasonable to speculate that the decrease in HBD-2, -3 and LL-37 could imply epithelial barrier dysfunction. The authors take into consideration the limitations of the study: a more quantitative approach (ELISA) would have provided more information and performing microbiome analysis would have considerably added depth to the study. 

Major comments: Of the 48 patients, how many had primary surgery and how many recurrent? Please state this in the text. Is there patient information of factors that might affect the outcome of different groups, such as patient allergies, asthma or smoking? How about demographics, such as age and sex? I available, a supplementary table could be included. Please address the possible effect in the text if such might exist.

The samples for the control group were taken from the inferior nasal turbinates. Is this anatomical site comparable to the site where samples were taken from the polyp patients, or is it possible that functional (immunological) differences exist, that might be reflected in the expression of the antimicrobial peptides? Please comment on this issue in the text. 

According to Table 1, the only molecule that has a different expression between primary and recurrent polyps in the connective tissue seems to be HBD-2. HBD-3 seems to be increased similarly in both surgery groups, compared to the control. The text on lines 181-183 states the same: "HBD-3 showed occasional number of positive structures (0/+) in both primary and recurrent nasal polyp epithelium and both demonstrated few to moderate positive structures (+/++) in subepithelial connective tissue (Figure 1c)". However, in the discussion, on lines 245-247 it says that "Our findings suggest an increased role of HBD-3 in patients with primary nasal polyps due to elevated HBD-3 concentration in subepithelial connective tissue in comparison to control samples and recurrent nasal polyp samples". This increase in HBD-3 in the connective tissue, when compared to the recurrent polyp group, is also mentioned in the abstract and conclusions as an implication of different tissue response to nasal microbiota between these two polyp patient groups. Which result is the correct one? If the expression of HBD-3 is elevated similarly in primary and recurrent polyp group, it should not be considered suggesting different tissue responses between the groups. 

Minor comment: Sentence on lines 281-283, "In one study HBD-2 was expressed in patients with primary in recurrent nasal polyps with rare expression in patients with objective recovery and control subjects, this analysis was done by RT-PCR that detected mRNA" is somewhat hard to understand, please modify. 

Author Response

Reviewer 2

Thank you for your valuable and constructive criticism. We addressed Your comments and suggestions point by point. Changes in the body of the manuscript are marked in green.

  1. Major comments: Of the 48 patients, how many had primary surgery and how many recurrent? Please state this in the text. Is there patient information of factors that might affect the outcome of different groups, such as patient allergies, asthma or smoking? How about demographics, such as age and sex? I available, a supplementary table could be included. Please address the possible effect in the text if such might exist.
  • Thank you, indeed the number of samples was missing. We have corrected this error:
    The study group was further divided into 29 patients undergoing their first surgery (patients with primary nasal polyps), 29 to 78 years old with average age of 48 years and 19 patients experiencing a recurrence of nasal polyps after previous surgeries (patients with recurrent nasal polyps), 31 to 88 years old with average age of 56 years. The control group consisted of 17 otherwise healthy individuals with an age range from 29 to 74 years with the average of 39 years with isolated nasal septum deviation and no known diagnosis of CRS or any other sinonasal pathologies.

Regarding comorbidities and other additional factors - this was not the aim of our study, but we addressed it in the limitations of the study and hope to include these parameters in future research.

  1. The samples for the control group were taken from the inferior nasal turbinates. Is this anatomical site comparable to the site where samples were taken from the polyp patients, or is it possible that functional (immunological) differences exist, that might be reflected in the expression of the antimicrobial peptides? Please comment on this issue in the text. 
  • Thank you, we have added comments on this issue:
    When comparing control samples to nasal polyps, anatomical site of sampling is a possible cause for discussion. Like stated before our control samples were taken from inferior turbinates of nasal cavity. Since respiratory part of nasal mucosa is lined by the same type of epithelium and subepithelial tissue structurally is the same as in ethmoidal region, mucosa of inferior nasal turbinates serves as a reference for normal nasal mucosa [45].

  1. According to Table 1, the only molecule that has a different expression between primary and recurrent polyps in the connective tissue seems to be HBD-2. HBD-3 seems to be increased similarly in both surgery groups, compared to the control. The text on lines 181-183 states the same: "HBD-3 showed occasional number of positive structures (0/+) in both primary and recurrent nasal polyp epithelium and both demonstrated few to moderate positive structures (+/++) in subepithelial connective tissue (Figure 1c)". However, in the discussion, on lines 245-247 it says that "Our findings suggest an increased role of HBD-3 in patients with primary nasal polyps due to elevated HBD-3 concentration in subepithelial connective tissue in comparison to control samples and recurrent nasal polyp samples". This increase in HBD-3 in the connective tissue, when compared to the recurrent polyp group, is also mentioned in the abstract and conclusions as an implication of different tissue response to nasal microbiota between these two polyp patient groups. Which result is the correct one? If the expression of HBD-3 is elevated similarly in primary and recurrent polyp group, it should not be considered suggesting different tissue responses between the groups. 
  • When describing results use pluses (+) for a visual representation. When values are transformed to numerical values for statistical analysis the results are depicted better and more understandable. We have updated table 1 with results expressed as numbers and p-values where it is shown that primary nasal polyp samples with HBD-3 positive structures in subepithelial connective tissue show a significantly different result when compared to control samples. Yet no significant changes are observed between HBD-3 expression in subepithelial connective tissue in primary and recurrent samples.
  1. Minor comment: Sentence on lines 281-283, "In one study HBD-2 was expressed in patients with primary in recurrent nasal polyps with rare expression in patients with objective recovery and control subjects, this analysis was done by RT-PCR that detected mRNA" is somewhat hard to understand, please modify. 
  • We have corrected the sentence to make more sense: In one study HBD-2 was expressed in patients with primary and recurrent nasal polyps but expression was rarely observed in patients with objective recovery and control subjects, this analysis was done by RT-PCR that detected mRNA [13].

Reviewer 3 Report

Congratulations. I think the design, development and results of your manuscript are worthy of its publication.

However, I miss some clarifications in the control group: 

. Any history of allergic rhinitis 

. Any previous history of chronic sinusitis 

. Any previous relevant nasal or nasosinus surgery. 

Would it be possible to clarify this fact?

Author Response

Reviewer 3

Thank you for your valuable and constructive criticism. We addressed Your comments and suggestions point by point. Changes in the body of the manuscript are marked in green.

I miss some clarifications in the control group: 

. Any history of allergic rhinitis 

. Any previous history of chronic sinusitis 

. Any previous relevant nasal or nasosinus surgery. 

Would it be possible to clarify this fact?

  • Thank you, we have clarified these facts in the body of the manuscript:
    The control group consisted of 17 otherwise healthy individuals with an age range from 29 to 74 years with the average of 39 years with isolated nasal septum deviation and no known diagnosis of CRS or any other sinonasal pathologies. Control group patients had no history of nasal surgery.

Round 2

Reviewer 1 Report

Despite the revision, the study cannot draw clear conclusions due to lack of critical clinical information (e.g. recent antibiotics use, allergic sensitization status). These parameters may be confounding factors for the results.

Specific points

The conclusion “Decrease of HBD-2 combined with an increase of HBD-3 in subepithelial connective tissue suggests different tissue responses to nasal microbiota in patients with primary nasal polyps compared to recurrent nasal polyps” seems incorrect. There was no significant difference in HBD-3 expression between primary nasal polyps and recurrent nasal polyps (Table 1).

=> Indeed, there was no significant difference between primary and recurrent nasal polyps in HBD-3 subepithelial connective tissue expression. Yet there was a significant difference between expression of HBD-3 in subepithelial connective tissue when comparing primary nasal polyp samples and the control group. The conclusion was driven by the latter.

- I cannot agree with the authors’ response. Because no significant difference in HBD-3 expression between primary nasal polyps and recurrent nasal polyps, the sentence “Decrease of HBD-2 combined with an increase of HBD-3 in subepithelial connective tissue suggests different tissue responses to nasal microbiota in patients with primary nasal polyps compared to recurrent nasal polyps” is definitely incorrect and should be revised.

Author Response

Reviewer 1

Thank you for your valuable and constructive criticism. We have addressed Your comments.

  1. Despite the revision, the study cannot draw clear conclusions due to lack of critical clinical information (e.g. recent antibiotics use, allergic sensitization status). These parameters may be confounding factors for the results.

  • As stated before, our aim was to characterize presence of antimicrobial peptides in tissue of nasal polyps. The presence of the factors that we chose to analyze in the tissue of nasal polyps and normal nasal mucosa do not belong to “fast reacting substances”, for instance, endocrine or diffuse neuroendocrine system substances, so their concentration in tissue cannot change quickly as a response to medication such as antibiotics. In our research of literature, we found no instances where use of medication could affect these factors as it takes a long time for their presence in tissue to change.
  • As for allergic sensitization – we agree that it could influence the overall state of tissue, yet antimicrobial peptides are not indicators of allergy and are not associated with an allergic process.
  • This study can be characterized as a form of basic medical research about factors involved in a specific disease. Tissues usually give similar reactions to various forms of irritation both from internal and external sources, so here we are describing specific tissue reactions of chronic rhinosinusitis with nasal polyps.
  • The conclusions that we are drawing are associated with morphological tissue changes, yet we agree that further analysis of clinical factors and comorbidities in the future could help bring these conclusions closer to practical medicine. We have addressed these points in the limitations of the study.

  1. I cannot agree with the authors’ response. Because no significant difference in HBD-3 expression between primary nasal polyps and recurrent nasal polyps, the sentence “Decrease of HBD-2 combined with an increase of HBD-3 in subepithelial connective tissue suggests different tissue responses to nasal microbiota in patients with primary nasal polyps compared to recurrent nasal polyps” is definitely incorrect and should be revised.

  • We have changed our conclusions accordingly: Decreased HBD-2, HBD-3 and LL 37 appearance in epithelium suggests a dysfunction of epithelial barrier in patients with nasal polyps. Decrease of subepithelial connective tissue HBD-2 suggests different responses to nasal microbiota in patients with primary nasal polyps compared to recurrent nasal polyps. Increase of HBD-3 in subepithelial connective tissue when compared to control samples suggests a possible role of this antimicrobial peptide in pathogenesis of primary nasal polyps.

Round 3

Reviewer 1 Report

The authors addressed all of my concerns.